# Temporal and Site-Specific ADP-Ribosylation Dynamics upon Different Genotoxic Stresses

**DOI:** 10.3390/cells10112927

**Published:** 2021-10-28

**Authors:** Sara C. Buch-Larsen, Alexandra K. L. F. S. Rebak, Ivo A. Hendriks, Michael L. Nielsen

**Affiliations:** Proteomics Program, Novo Nordisk Foundation Center for Protein Research, Faculty of Health and Medical Sciences, University of Copenhagen, Blegdamsvej 3B, 2200 Copenhagen, Denmark; sara.larsen@cpr.ku.dk (S.C.B.-L.); alexandra.stripp@cpr.ku.dk (A.K.L.F.S.R.); ivo.hendriks@cpr.ku.dk (I.A.H.)

**Keywords:** ADP-ribosylation, PARP, DNA damage, Af1521, post-translational modification, proteomics, mass spectrometry

## Abstract

The DNA damage response revolves around transmission of information via post-translational modifications, including reversible protein ADP-ribosylation. Here, we applied a mass-spectrometry-based Af1521 enrichment technology for the identification and quantification of ADP-ribosylation sites as a function of various DNA damage stimuli and time. In total, we detected 1681 ADP-ribosylation sites residing on 716 proteins in U2OS cells and determined their temporal dynamics after exposure to the genotoxins H_2_O_2_ and MMS. Intriguingly, we observed a widespread but low-abundance serine ADP-ribosylation response at the earliest time point, with later time points centered on increased modification of the same sites. This suggests that early serine ADP-ribosylation events may serve as a platform for an integrated signal response. While treatment with H_2_O_2_ and MMS induced homogenous ADP-ribosylation responses, we observed temporal differences in the ADP-ribosylation site abundances. Exposure to MMS-induced alkylating stress induced the strongest ADP-ribosylome response after 30 min, prominently modifying proteins involved in RNA processing, whereas in response to H_2_O_2_-induced oxidative stress ADP-ribosylation peaked after 60 min, mainly modifying proteins involved in DNA damage pathways. Collectively, the dynamic ADP-ribosylome presented here provides a valuable insight into the temporal cellular regulation of ADP-ribosylation in response to DNA damage.

## 1. Introduction

ADP-ribosylation (ADPr) is an emerging post-translational modification (PTM), involved in a variety of cellular processes including DNA repair [1,2,3]. The modification is catalyzed by a group of enzymes called ADP-ribosyltransferases (ARTs) that use NAD^+^ to modify target proteins either with one ADP-ribose moiety (mono-ADPr, MARylation) or with sequential addition of multiple moieties (poly-ADPr, PARylation) [2,4]. Although PARylation is believed to account for the majority of nuclear ADPr, recent studies have questioned how much of the ADPr is actually present as PARylation [5,6]. These observations could be explained by ADPr being a reversible and highly transient PTM, with different hydrolases able to hydrolyze the glycosidic bonds between poly-ADPr moieties or between the ADPr moieties that are attached to target amino acid residues [7,8,9,10,11]. Of the ARTs, PARP1 is historically the best characterized, due to its important role in DNA damage repair [12]. Here, PARP1 has proved important in single-strand break repair (SSBR), double-strand break repair (DSBR), and stabilization of replication forks [3]. As a result, several PARP1 inhibitors are widely used in clinical settings for the treatment of breast cancer and ovarian cancer [13,14,15]. Traditionally, ADPr has primarily been described as occurring on glutamic acid residues and aspartic acid residues [16,17,18]; however, the modification can target chemically distinct amino acids including cysteine, histidine, lysine, arginine, serine, threonine, and tyrosine residues [19,20]. In recent years, serine residues have emerged as the primary target for PARP1- and PARP2-mediated ADPr, especially in the DNA damage response (DDR) when the co-factor histone PARylation factor 1 (HPF1) is engaged [19,21,22,23,24,25,26].

Mass spectrometry (MS)-based proteomics has emerged as a highly valuable technology for global and unbiased analyses of PTMs [27,28,29,30]. In the past decade, substantial efforts have been made towards developing MS-based strategies for studying ADPr at the system-wide level [18,19,23,24,31,32,33,34,35,36]. As most MS-based studies have focused on method development, the ADP-ribosylome has primarily been studied in the context of oxidative stress (H_2_O_2_ treatment), since H_2_O_2_ is known to induce a strong ADPr signal response [7,19,23,24,31,32,35,37,38]. However, H_2_O_2_ does not constitute the most specific genotoxic stress-inducing agent, as relatively high concentrations can activate both the base excision repair (BER) pathway through the oxidation of bases and the SSBR pathway or the DSBR pathway through breaks generated in the backbone [39,40,41]. Only a few system-wide studies have investigated the influence of different genotoxic stresses on the ADP-ribosylome, including methyl methanesulfonate (MMS) [31], methyl nitro-nitrosoguanidine (MNNG) [18,33,42], ultraviolet (UV) radiation [31], and ionizing radiation (IR) [31]. While these studies provide valuable insights into which proteins are ADPr-modified in response to DNA damage, they have been hampered by limitations, e.g., by only investigating ADPr-modified proteins instead of modification sites or only being able to identify ADPr occurring on aspartic acid and glutamic acid residues. Here, we utilize our unbiased MS-based methodology for system-wide and site-specific enrichment of ADPr [19,23,24], and investigate the temporal ADPr dynamics upon oxidative (H_2_O_2_) and alkylating (MMS) stress.

## 2. Materials and Methods

### 2.1. Cell Culture and Lysis

U2OS cells (HTB-96) and HeLa cells (CCL-2) were acquired via the American Type Culture Collection (ATCC) and cultured in Dulbecco’s modified Eagle’s medium (Thermo Fisher Scientific Invitrogen, Waltham, MA, USA) supplemented with 10% of fetal bovine serum (FBS) and penicillin/streptomycin (100 U/mL; Gibco) at 37 °C and 5% CO_2_. Cells were routinely tested for mycoplasma contamination. For the immunoblot experiments, the U2OS cells and HeLa cells were treated with hydrogen peroxide (H_2_O_2_; 1, 2, 5, or 10 mM, Sigma Aldrich), hydroxy urea (HU; 1 or 10 mM, Sigma Aldrich, St. Louis, MO, USA), mitomycin C (MMC; 0.01 or 0.1 mM, Sigma Aldrich), methyl methanesulfonate (MMS; 0.5 or 5 mM, Sigma Aldrich), cisplatin (0.5 or 5 mM, Sigma Aldrich), or neocarzinostatin (NCS; 5 or 50 nM, Sigma Aldrich) for 10 min at 37 °C. For the mass spectrometry (MS) experiments, the U2OS cells were treated with 5 mM H_2_O_2_ or 5 mM MMS for 1, 10, 30, or 60 min. Untreated cells were used as a control. For the immunoblot experiments, the U2OS cells and HeLa cells were lysed in STBS buffer (2% SDS, 150 mM NaCl, 50 mM TRIS–HCl, pH 8.5) at room temperature, and were homogenized by shaking at 99 °C for 30 min. For the MS experiments, cells were washed twice with ice-cold phosphate-buffered saline (PBS) and collected by gentle scraping at 4 °C. Cells were pelleted by centrifugation at 4 °C for 3 min at 500× *g*. The PBS was decanted, and the cell pellets were directly lysed in 10 pellet volumes of lysis buffer (6 M guanidine-HCl, 50 mM TRIS, pH 8.5) followed by vigorous vortexing and shaking of the samples. Lysates were snap frozen using liquid nitrogen and stored at −80 °C for further processing.

### 2.2. Immunoblot Analysis

Protein concentrations were determined using Bradford reagent (Bio-Rad, Hercules, CA, USA). Prior to loading, lysates were supplemented with NuPAGE LDS sample buffer (Invitrogen) and dithiothreitol (DTT) and separated on 4–12% Bis-Tris gels using MOPS running buffer. Proteins were transferred to Amersham™ Protran^®^ nitrocellulose membranes (GE Healthcare, Chicago, IL, USA), and the membranes were blocked using 5% BSA solution in PBS supplemented with 0.1% of Tween 20 (PBST) for 1 h. Afterwards, membranes were incubated with poly/mono-ADP-ribose (E6F6A) rabbit mAb #83732 (CST) 1:1000 with overnight rotation at 4 °C, and afterwards washed three times with PBST. Membranes were incubated with goat-anti-rabbit HRP conjugated secondary antibody (111-036-045, Jackson ImmunoResearch, West Grove, PA, USA), at a concentration of 1:10,000 for 1 h with shaking at room temperature. Membranes were washed three times with PBST prior to detection using a Novex ECL Chemiluminescent Substrate Reagent Kit (Invitrogen).

### 2.3. Protein Digestion and Sample Cleanup

Lysates were thawed at room temperature and homogenized using sonication. Subsequently, samples were reduced and alkylated using 5 mM tris(2-carboxyethyl) phosphine (TCEP) and 5 mM chloroacetamide (CAA) for 1 h at 30 °C. Samples were first digested with lysyl endopeptidase (Lys-C, 1:100 *w/w*; Wako Chemicals, Richmond, VA, USA) for 3 h at room temperature, then diluted with three volumes of 50 mM ammonium bicarbonate (ABC), after which they were further digested using modified sequencing grade trypsin (1:100 *w/w*; Sigma Aldrich) overnight at room temperature. The resulting peptide mixtures were acidified by addition of trifluoroacetic acid (TFA) to a final concentration of 0.5% (*v/v*) and cleared by centrifugation, and the peptides were purified using reversed-phase C18 cartridges (Sep-Pak, Waters, Milford, MA, USA). Elution of peptides was performed with 30% acetonitrile (ACN) in 0.1% TFA, and peptides were frozen overnight at −80 °C and afterwards lyophilized for 96 h. Lyophilized peptides were stored at −80 °C prior to further processing.

### 2.4. Purification of ADP-Ribosylated Peptides

ADP-ribosylated peptides were enriched essentially as described previously [19,23,24]. In brief, lyophilized peptides were dissolved in AP buffer (50 mM TRIS, pH 8.0, 1 mM MgCl_2_, 250 μM DTT, and 50 mM NaCl), and cleared by centrifugation at room temperature for 30 min at 4250× *g*. The peptide concentration was determined using a NanoDrop™ 2000/2000c spectrophotometer, and ~10 mg of peptide was used for each replicate experiment. Any ADP-ribose polymers were reduced to monomers by incubation with poly (ADP-ribose) glycohydrolase (PARG, a kind gift from Professor Michael O. Hottiger) at a concentration of 1:10,000 (*w/w*) with overnight shaking at room temperature. Following PARG treatment, samples were cleared by centrifugation at 4 °C for 30 min at 4250× *g* and transferred to new 15 mL tubes. Subsequently, 100 µL of sepharose beads coated with in-house-produced GST-tagged Af1521 macrodomain [19,23,24,31,32] was added to each sample on ice, and samples were incubated with head-over-tail rotation at 4 °C for 4 h. The beads were washed twice in freshly prepared ice-cold AP buffer, twice in ice-cold PBS with DTT, and twice in ice-cold MQ water, with a tube change every time the buffer was changed. ADPr-modified peptides were eluted off the beads by the addition of ice-cold 0.15% TFA and incubated twice for 20 min, with mixing every 5 min. Eluted peptides were passed through 0.45 μm spin filters, and afterwards through pre-washed 100 kDa cut-off spin filters (Vivacon 500, Sartorius, Göttingen, Germany), after which they were stored at −80 °C prior to further processing.

### 2.5. Fractionation of ADP-Ribosylated Peptides

ADPr-modified peptides were fractionated on StageTips at high pH, essentially as described previously [19,23,24,43]. In brief, StageTips containing four layers of C18 disc material were prepared, activated with methanol and 80% ACN in 50 mM ammonium hydroxide, and equilibrated twice with 50 mM ammonium hydroxide [24,44]. Prior to loading, samples were basified by addition of ammonium hydroxide to a final concentration of 20 mM. After sample loading, the StageTips were washed twice with 50 mM ammonium hydroxide, and the peptides were eluted as four fractions (F1–F4) by increasing the amount of ACN in 50 mM ammonium hydroxide. The flow-throughs from loading the sample and the first wash were pooled and acidified, after which the remaining peptides were purified via StageTips at low pH and eluted as fraction 0 (F0). All samples were dried to completeness in a SpeedVac at 60 °C, and the peptides were afterwards dissolved in a few μL of 0.1% formic acid (FA). Tubes containing dissolved peptides were gently tapped, spun down, and stored at −20 °C prior to mass spectrometric measurements.

### 2.6. Mass Spectrometric Analysis

All MS experiments were analyzed on an EASY-nLC 1200 HPLC system (ThermoFisher Scientific, Waltham, MA, USA) connected to a Orbitrap Fusion Lumos mass spectrometer (Thermo) equipped with a Nanospray Flex Ion Source (ThermoFisher Scientific). Each sample was separated in a 15 cm analytical column with an internal diameter of 75 μm, packed in house with 1.9 μm C18 beads (ReproSil-Pur AQ, Dr. Maisch, Ammerbuch, Germany), and heated to 40 °C using a column oven. Peptide separation was performed using a 60 min gradient at a flow rate of 250 nL/min, utilizing buffer A consisting of 0.1% of FA, and buffer B consisting of 80% of ACN in 0.1% of FA. The primary gradient ranged from 3% buffer B to 24% buffer B over 37 min, followed by an increase to 40% buffer B over 9 min to ensure elution of all peptides, followed by a washing block of 14 min. The effluent from the HPLC was directly electrosprayed into the mass spectrometer. The spray voltage was set to 2 kV, the capillary temperature to 275 °C, and the RF level to 30%. The Fusion Lumos mass spectrometer was operated in data-dependent acquisition mode, with full scans performed at a resolution of 120,000, a scan range of 300 to 1750 *m*/*z*, a maximum injection time of 250 ms, and an automatic gain control (AGC) target of 600,000 charges. Precursors were isolated with a width of 1.3 *m*/*z*, with an AGC target of 200,000 charges, and precursor fragmentation was accomplished using electron transfer disassociation with supplemental higher-collisional disassociation (EThcD), with supplemental activation energy of 20. Precursors with charge state 3–5 were included, prioritized from charge 3 (highest) to charge 5 (lowest), using the decision tree algorithm. Selected precursors were excluded from repeated sequencing by setting a dynamic exclusion of 60 s. MS/MS spectra were measured in the Orbitrap, with a loop count setting of 3, a maximum precursor injection time of 500 ms, and a scan resolution of 60,000.

### 2.7. Data Analysis

All MS raw data were analyzed using the MaxQuant software suite version 1.5.3.30 [45], and searched against the human proteome in FASTA file format, as downloaded from UniProt on the 24 May 2019. The default MaxQuant settings were used except that protein N-terminal acetylation (default), methionine oxidation (default), cysteine carbamidomethylation, and ADP-ribosylation on cysteine, aspartic acid, glutamic acid, histidine, lysine, arginine, serine, threonine, and tyrosine residues were included as variable modifications. A maximum of 4 variable modifications and 6 missed cleavages were allowed. Matching between runs was enabled with a match time of 1 min and an alignment time window of 20 min. The Andromeda delta score was set to a minimum of 20 for modified peptides. The data were automatically filtered by MaxQuant to obtain a false discovery rate (FDR) of less than 1% at the peptide-spectrum match (PSM) level (default), the protein assignment level (default), and the site-specific level (default), and the data were additionally manually filtered in order to ensure the proper identification and localization of ADP-ribose. PSMs corresponding to unique modified peptides were only used for ADP-ribosylation site assignment if the localization probability was >0.90, although localization of >0.75 was accepted for purposes of intensity assignment of further evidence for unique peptides already localized with at least one > 0.90 evidence. PSMs harboring two or more ADPr moieties, single-site evidences erroneously assigned to multiple sites by MaxQuant, PSMs matching the reversed-sequence database used for FDR control, and PSMs matching non-human contaminant proteins were all omitted from further analysis. In cases of ambiguity where peptides could be assigned to multiple protein isoforms, we assigned peptides to the most canonical and well-annotated UniProt identifier, for listing in the final data tables (Appendix A).

## 3. Results

### 3.1. H_2_O_2_ and MMS Induces a Strong ADP-Ribosylation Response

Catalytic activation of PARP1 or PARP2 is the first step in nuclear ADPr signaling, which extends through a cascade of downstream substrates to mediate ADP-ribosylation of a large number of substrates [24,26,46]. The overall levels of ADPr-modified proteins within the nucleus, however, are also affected by other processes, including removal of ADPr by hydrolases [7,8,9,10,11], protein translocation [47] and protein turnover [48]. Nonetheless, our method measures the net effect of all these diverse processes that collectively regulate the dynamic ADP-ribosylome. The ADPr signaling response upon H_2_O_2_ treatment has been extensively studied using mass spectrometry (MS)-based strategies [19,23,24,36,49]; however, knowledge about the temporal aspects of ADPr signaling upon different DNA-damage-inducing agents is still limited. Consequently, we initially investigated the effect of six different genotoxic stress-inducing agents on the ADPr-ribosylome in both HeLa and U2OS cells, using immunoblot analysis. To this end, we compared untreated cells to cells treated for 10 min with various concentrations of either hydrogen peroxide (H_2_O_2_), hydroxyl urea (HU), mitomycin C (MMC), methyl methanesulfonate (MMS), cisplatin (Cis), or neocarzinostatin (NCS), and visualized the ADPr signal with immunoblot analysis using a commercially available poly/mono-ADPr antibody from Cell Signaling Technology (Figure 1A and Appendix A). A strong ADPr signal corresponding to modified histones and a clear band related to the auto-modified PARP1 were observed across all investigated conditions. Notably, the strongest induction of ADPr was observed for H_2_O_2_-treated cells, followed by cells treated with MMS. Intriguingly, the MMS-driven induction of ADPr was considerably higher in U2OS cells compared to HeLa cells.

Based on the observations from our immunoblot analysis, we decided to perform MS-based experiments aimed at investigating the temporal response to H_2_O_2_ and MMS treatment. To this end, we prepared quadruplicate cell cultures of mock-treated U2OS cells and U2OS cells treated with 5 mM H_2_O_2_ or 5 mM MMS, and lysed the cells after 1, 10, 30, or 60 min (Figure 1B). Following digestion, the peptides were enriched using our unbiased Af1521 enrichment strategy [19,23,24,32], prior to high-pH fractionation using StageTips [43]. ADPr-enriched samples were analyzed on a Fusion Lumos mass spectrometer using EThcD fragmentation to pinpoint the exact ADPr acceptor sites [19,23,24], followed by processing using the MaxQuant software [45]. In total, we identified 1681 confidently localized ADPr sites (Appendix A) residing on 716 proteins (Appendix A), with the majority of identified proteins being ADP-ribosylated on either one or two acceptor sites, with an average of 2.4 ADPr sites per protein (Appendix A). The largest number of confidently localized ADPr sites was observed after MMS treatment for 30 min, closely followed by treatment with H_2_O_2_ for 30 min (Figure 1C). As expected, the lowest number of ADPr sites was observed in the control conditions, where we nonetheless succeeded in identifying 268 physiological modification sites in untreated cells. Overall, we found that the ADPr abundance correlated well with the number of ADPr sites identified (Figure 1D). Intriguingly, after just 1 min of treatment, we had already identified ~300 ADPr sites on average, albeit with low abundances, comparable to the intensities observed for the control conditions. In our immunoblot analysis we noticed a strong signal from ADPr-modified histones (Figure 1A), and our MS analysis confirmed that histones were heavily ADPr-modified (Appendix A), with >70% of the total ADPr signal originating from modified histones across all investigated conditions. Intriguingly, untreated cells and cells treated with MMS for 60 min showed the highest fraction of histone ADPr, with more than 80% of the total ADPr signal coming from modified histones. This supports the idea that histones generally are a major cellular target of nuclear ADPr.

Overall, we obtained confident localization of ADPr within modified peptides, with approximately 70% of all ADPr peptide-spectrum matches (PSMs) showing a localization probability above 0.90 (Appendix A), and 82% showing a localization probability above 0.75. Generally, we found that same-condition replicates clustered well, and we noticed a strong tendency for the ADP-ribosylomes induced by the two damage-inducing agents to resemble each other (Figure 1E,F), with the temporal aspect inducing the most variance. Previously, DNA-damage-induced ADPr has been described as mainly occurring on serine residues in the presence of HPF1 [19,21,22,24,25], and similarly, we found that ADPr-modified serine residues accounted for the vast majority (>90%) of the relative ADPr abundance regardless of DNA damage treatment and time points (Appendix A). Additionally, we confirmed the presence of the ADPr-specific KS motif [21,24] (Figure 1G), with more than 70% of the relative serine-ADPr abundance occurring in KS motifs (Appendix A). Moreover, we noticed the prevalence of an SGG motif, which we have previously described as being driven by abundance bias [19].

In summary, we demonstrated that the investigated genotoxins H_2_O_2_ and MMS both induce a robust ADPr response, and with both treatments we found that the strongest induction of ADPr was achieved after 30 min. Additionally, we observed that modified histones accounted for the majority of the total ADPr signal, and we confirmed that serine residues are the predominant target of ADPr across all investigated conditions.

### 3.2. The ADP-Ribosylome Is Homogenous upon H_2_O_2_- and MMS-Treatment

In total, we identified 1583 and 1612 ADPr sites residing on 676 and 687 ADPr-modified proteins upon H_2_O_2_ and MMS treatment, respectively (Appendix A). Intriguingly, the size and magnitude of the ADP-ribosylomes induced by H_2_O_2_ and MMS correlated well (Appendix A), and generally we did not observe any significant difference between the ADPr-modified proteins identified upon these two distinct types of damage (Appendix A). Similarly, we did not observe any differences in the amino acid distributions between ADPr sites identified upon H_2_O_2_ (Figure 2A, middle panel) and MMS treatment (Figure 2A, right panel), with serine residues accounting for more than 90% of the number of identified ADPr sites (Figure 2A, top panel) and close to 100% of the ADPr abundance (Figure 2A, bottom panel). For untreated cells, serine residues constituted a smaller fraction compared to stressed cells (84% of ADPr sites and 97% of ADPr abundance, Figure 2A, left panel), but nevertheless serine residues remained the main acceptor sites. The remaining ADPr signal was observed mainly on arginine residues, lysine residues, and histidine residues (Appendix A). The majority (>80%) of identified ADPr-modified proteins were annotated as either nuclear-specific or as both nuclear and cytoplasmic (Figure 2B), and this distribution likewise did not notably change between the different conditions, supporting the idea that the identified ADPr sites and ADP-ribosylated proteins are likely targets of active nuclear PARP enzymes.

To the best of our knowledge, only one other study has compared different genotoxins in a system-wide manner [31], and when comparing the ADPr-modified proteins upon MMS treatment identified in their study to the ones we identified, we observed a notable overlap, and further expanded the stress-induced ADP-ribosylome (Appendix A).

Overall, we observed a substantial overlap of the individual ADPr sites (Figure 2C) and proteins (Figure 2) identified by both treatments, with practically all modified sites and proteins identified in the control conditions also present under stimulated conditions. Compared to the control conditions, and while concomitantly considering all stresses at all time points, we observed 154 proteins to be significantly enriched upon genotoxic stress (Appendix A), supporting the idea that these proteins are significantly regulated under all tested conditions and thus are likely to represent major global targets of ADPr. These common damage-induced ADPr-modified proteins identified in response to both H_2_O_2_ and MMS damage, were observed to be functionally highly interconnected and related to RNA processing, chromosome organization, and cellular response to DNA damage stimuli (Figure 2E).

Taken together, we find the ADP-ribosylomes induced by H_2_O_2_ or MMS to be surprisingly homogenous with regard to both ADPr acceptor sites and ADPr-modified proteins. 

### 3.3. Temporal Profiles of the ADP-Ribosylome

Whereas the effect of H_2_O_2_ on ADPr has been extensively studied at the system-wide level [19,23,24,36,49], less is known about the temporal regulation of the site-specific ADP-ribosylome [32]. To address this, we investigated the ADPr response after 1, 10, 30, and 60 min of treatment with H_2_O_2_ and MMS. Reassuringly, the various time points revealed numerous significantly regulated ADPr-modified proteins upon genotoxic stress, compared to the control conditions (Appendix A). As previously mentioned, both types of damage induced a gradual increase in the total number of sites identified as well as the summed ADPr abundance, with 1 min of damage resulting in the fewest regulated proteins, peaking after 30 min of treatment and declining after 60 min of treatment. This decline in the ADPr response was more pronounced for MMS compared to H_2_O_2_ treatment (Figure 3A,B). Intriguingly, the increase in overall ADPr abundance occurred slightly later than the numerical increase in ADPr sites, suggesting that a widespread distribution of ADPr occurs early in the process but at low abundance, and is followed by further modification of additional proteins at the same sites, leading to an overall amplification of ADPr abundance.

PARP1 is known to be one of the first responders to DNA damage, where it auto-modifies itself upon binding to DNA breaks [50,51]. Previously, three serine residues, S-499, S-507, and S-519, have been described as the main targets of auto-modification on PARP1 [19,25,52], and we were able to confirm these sites as being the most abundantly modified on PARP1 in this study (Appendix A). Notably, we found S-499 to be the most abundant auto-modification site on PARP1 in untreated cells or cells treated at the earliest time point (1 min) with either H_2_O_2_ or MMS, whereas S-507 became the most abundantly modified site on PARP1 when treated with genotoxic stress for a longer time. In contrast, the abundance of S-519 remained low across all investigated conditions. S-504 was identified as the fourth most intense auto-modification site, and showed the same tendency as S-519, but with a 10-times-lower intensity. With the individual auto-modification sites on PARP1 revealing interesting dynamics, we next investigated the overall PARP1 modification abundance. Intriguingly, for both H_2_O_2_ and MMS, we observed that the abundance of auto-modified PARP1 peaked between 10 and 60 min (Figure 3C, blue line), in spite of the ability of PARP1 to respond to DNA damage within seconds [53]. To investigate whether other ADP-ribosylated proteins exhibited similar trends, we extracted the 15 proteins demonstrating the temporal profiles most similar to PARP1 (Figure 3C,D, red lines). Several of these ADPr-modified proteins corresponded to known players in the DDR, for example HP1BP3 [54] and FEN1 [55]. To gain more insight into the connectivity, we expanded the analysis to include the 50 proteins with profiles most similar to PARP1 and found a strong connectivity and an enrichment of RNA metabolic processes, chromosome organization, and DNA repair (Figure 3C,E, orange lines).

In summary, we showed that H_2_O_2_ and MMS generated similar temporal profiles, with the accumulation of ADPr-modified proteins in response to both genotoxic stresses peaking surprisingly late in U2OS cells, and we found that proteins with temporal modification profiles similar to PARP1 are generally involved in DNA repair, chromosome organization, and RNA metabolic processes. 

### 3.4. Temporal-Specific Changes in ADP-Ribosylation Dynamics by H_2_O_2_ and MMS

Despite both genotoxic stresses generating a homologous ADPr response and showing overall similar temporal profiles, we next explored potential abundance differences in the ADP-ribosylated proteins across the individual time points. A comparison of the H_2_O_2_- and MMS-induced ADPr response revealed no significant differences after 1 min of treatment (Appendix A) and only a few significantly regulated ADPr-modified proteins after 10 min of treatment (Appendix A). In contrast, we observed 207 ADPr-modified proteins to be significantly regulated after 30 min of stress, and the majority of these were more abundantly ADP-ribosylated in response to the MMS treatment (Figure 4A). After 60 min of stress, we observed 285 ADPr-modified proteins to be significantly upregulated, but with the majority now more abundantly ADP-ribosylated in response to H_2_O_2_ treatment (Figure 4B). We were intrigued by the observed shift in the ADP-ribosylome from being MMS-driven at 30 min of treatment to H_2_O_2_-driven at 60 min of treatment, and we wondered whether the same proteins were differentially regulated by the two stresses at these distinct time points. Therefore, we compared the proteins significantly upregulated after 30 min of MMS treatment to the proteins significantly upregulated after 60 min of H_2_O_2_ treatment. Whereas we observed a notable overlap (20%) between the two groups of differentially regulated proteins, the majority of proteins were found to be specific for either of the groups (Figure 4C, top panel). We found 110 ADPr target proteins upregulated after 30 min of MMS treatment, with functions related to rRNA processing, transcription, translation, and cell cycle progression (Figure 4C, bottom left panel). Treatment with H_2_O_2_ for 60 min resulted in the highest number of specifically upregulated proteins (197), and interestingly, these were also found to be involved in rRNA processing, as well as DNA repair, including nucleotide-excision repair and base-excision repair (Figure 4C, bottom right panel).

Overall, we showed that even though cellular treatment with H_2_O_2_ and MMS induced similar temporal ADP-ribosylome profiles, we nonetheless observed significant and genotoxin-specific differences at distinct time points. MMS treatment resulted in the highest number of highly ADP-ribosylated proteins after 30 min of treatment, whereas after 60 min the highest prevalence of ADP-ribosylation was observed in response to H_2_O_2_ treatment. Notably, MMS-specific ADPr target proteins were more frequently involved in transcriptional regulation and cell cycle progression, whereas H_2_O_2_-specific ADPr target proteins modulated the DNA damage response. In conclusion, while various genotoxic insults may initiate a similarly homogenous and robust ADPr response, the final form of the reactive ADP-ribosylome could be subtly yet distinctly unique for each cellular stress.

## 4. Discussion

In this study, we investigated the induction of ADP-ribosylation upon a range of genotoxic stresses and, using immunoblotting, we found that H_2_O_2_ and MMS were the strongest perpetuators of ADP-ribosylation. To explore the ADPr signaling response at the molecular level, we utilized our Af1521-based proteomics method for unbiased enrichment of ADPr-modified peptides and explored the temporal aspects of the ADP-ribosylome upon H_2_O_2_ and MMS treatment. The effects of H_2_O_2_ on the ADPr equilibrium have been extensively studied [7,19,23,24,31,32,35,37,38], and different levels of oxidative stress have been shown not to affect the major part of the ADP-ribosylome [37]. However, to the best of our knowledge, only one study has investigated the system-wide effect of the alkylating agent MMS on ADPr [31]. The authors found that H_2_O_2_ induced the strongest regulation of the ADP-ribosylome, followed by MMS treatment. However, it should be noted that the study by Jungmichel and colleagues used a slightly higher concentration of MMS (10 mM), and that they used a protein-level enrichment strategy, not in a site-specific manner. Whereas this allowed identification of the ADPr-modified proteins, it did not profile the exact residues modified by ADPr; moreover, protein-level enrichment strategies often suffer from false-positive identification via non-specific non-covalent interactions. Nevertheless, we confirmed that H_2_O_2_ and MMS treatment induced the strongest ADPr response of all genotoxins tested, and that both H_2_O_2_ and MMS resulted in ADP-ribosylation of proteins involved in RNA metabolic processes.

In total, we reported 1681 confidently localized ADPr sites residing on 716 proteins, with the large majority of these proteins annotated as nuclear. We confirmed recent findings demonstrating that serine residues are the primary target of ADPr upon DNA damage when the cofactor HPF1 is present [21,22,25,26]. Additionally, we found that the majority of ADPr resides on serine residues under physiological conditions, as we recently demonstrated [23]. Moreover, we found that more than 70% of identified serine residues occurred in the previously reported KS motifs [19,21,23,24,25,35] across all investigated conditions.

Overall, we found that H_2_O_2_ and MMS induced a comparatively homogenous ADP-ribosylome, with few ADPr sites and ADPr-modified proteins specific to either of the two genotoxic stresses. Reassuringly, we found proteins upregulated by both stresses to be involved in known ADPr biological processes such as RNA processing, chromosome organization, and the response to DNA damage stimuli. In the ADPr research field, cells are often exposed to H_2_O_2_ for 10 min [19,23,24,25,26,31,37], while few system-wide studies have examined the temporal effects [32]. Here, we performed a system-wide investigation of the temporal effects on the ADP-ribosylome of H_2_O_2_ and MMS by measuring ADPr-modified peptides after 1, 10, 30, and 60 min of treatment. We found the highest number of identified ADPr sites and the highest overall ADPr abundance after 30 min of treatment for both H_2_O_2_ and MMS. This is in contrast with a previous study where the ADPr signal peaked after 5 to 10 min in HeLa cells, as determined by both microscopy and MS analysis [32] and with the general notion in the field that ADPr is generated swiftly upon damage, and then rapidly turned over. Likewise, we observed that both H_2_O_2_ and MMS treatment induced the highest levels of auto-modified PARP1 between 10 and 60 min of treatment, although PARP1 is known to be one of the first responders to DNA damage [50,51,53]. However, PARP1 auto-modification is not a direct measure of DNA repair, as PARP1 auto-modification causes PARP1 to be evicted from the DNA [56,57]. We confirmed S-499, S-507, and S-519 as the main auto-modification sites on PARP1 and, intriguingly, we observed a shift from S-499 being the most abundant site in untreated or briefly treated cells to S-507 being the most abundant site when cells were treated for longer times.

Despite the globally similar temporal profiles observed for H_2_O_2_ and MMS, we did observe subtle differences at specific time points. Intriguingly, at 30 min, MMS treatment resulted in a relative upregulation of proteins involved in rRNA processing, transcription, and translation. In contrast, at 60 min, H_2_O_2_ treatment resulted in a relative upregulation of proteins involved in rRNA processing and DNA repair.

In summary, we presented a damage-specific and temporal ADP-ribosylome, which constitutes an important biological resource and provides important insights into ADPr induced in response to oxidative and alkylating stress.

## Figures and Tables

**Figure 1 cells-10-02927-f001:**
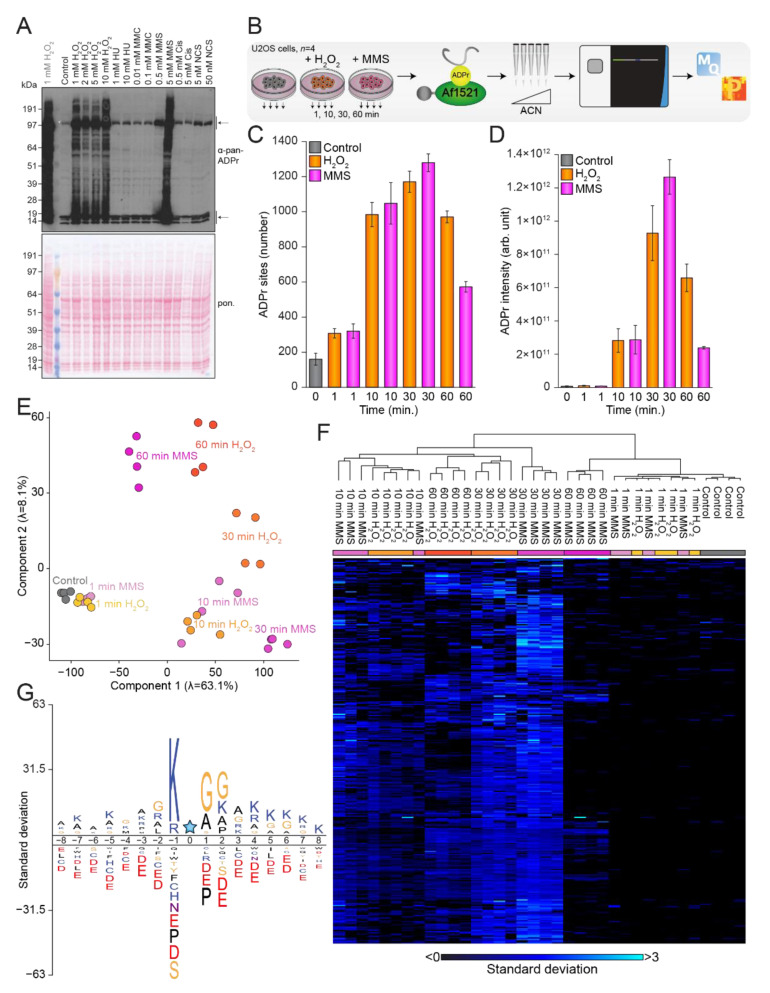
Induction of the ADPr signaling response. (**A**) Immunoblot analysis illustrating the ADPr equilibrium upon treatment with various concentrations of hydrogen peroxide (H_2_O_2_), hydroxyurea (HU), mitomycin C (MMC), methyl methanesulfonate (MMS), cisplatin (Cis), and neocarzinostatin (NCS) in U2OS cells. The top arrow indicates auto-modified PARP1, and the bottom arrow indicates histone ADPr. The left lane (grey text) corresponds to lane 2 in Appendix A. (**B**) Overview of the experimental design. U2OS cells were left untreated or treated with 5 mM H_2_O_2_ or 5 mM MMS for 1, 10, 30, or 60 min in quadruplicate, followed by enrichment of ADPr-modified peptides using the Af1521 macrodomain. Samples were fractionated at high pH, analyzed on the Fusion Lumos mass spectrometer, and processed using MaxQuant. (**C**) Histogram showing the number of ADPr sites identified. With *n* = 4 cell culture replicates, data are presented as mean values +/− standard deviations. (**D**) As **C** but showing the ADPr intensity. With *n* = 4 cell culture replicates, data are presented as mean values +/− standard errors of the mean. (**E**) Principal component analysis indicating the highest degree of variance. (**F**) Hierarchical clustering analysis based on z-scored log_2_-transformed suADPr site intensities. (**G**) The iceLogo analysis showing the sequence context surrounding identified serine ADPr sites (blue star), with amino acid residues above the line being enriched (*p* < 0.01). Sequence windows from all serine residues in ADPr target proteins were used as a reference.

**Figure 2 cells-10-02927-f002:**
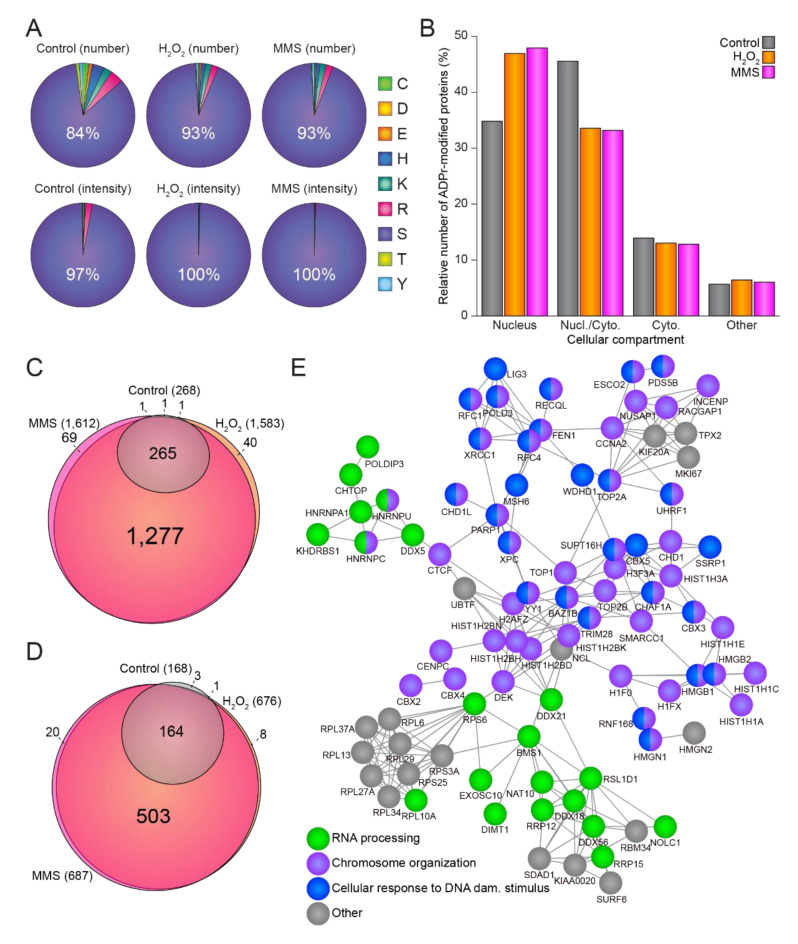
Damage-specific properties of the ADP-ribosylome. (**A**) Pie charts illustrating the amino acid distribution of the identified ADPr sites (top panel) or ADPr intensity (lower panel) under control conditions (left panel), H_2_O_2_ treatment (middle panel), or MMS treatment (right panel). (**B**) Histogram depicting the distribution of ADPr-modified proteins in different cellular compartments. (**C**) Scaled Venn diagram showing the overlap of confidently localized ADPr sites between control, H_2_O_2_ treatment, and MMS treatment. (**D**) As C, but illustrating the overlap of the ADPr-modified proteins. (**E**) STRING network showing interactions between proteins significantly enriched after either H_2_O_2_ or MMS treatment compared to control conditions. Default STRING clustering was used (*p* > 0.4) except for disabling of text mining, and disconnected proteins were omitted from the network. Proteins were annotated with colors as highlighted in the figure legend.

**Figure 3 cells-10-02927-f003:**
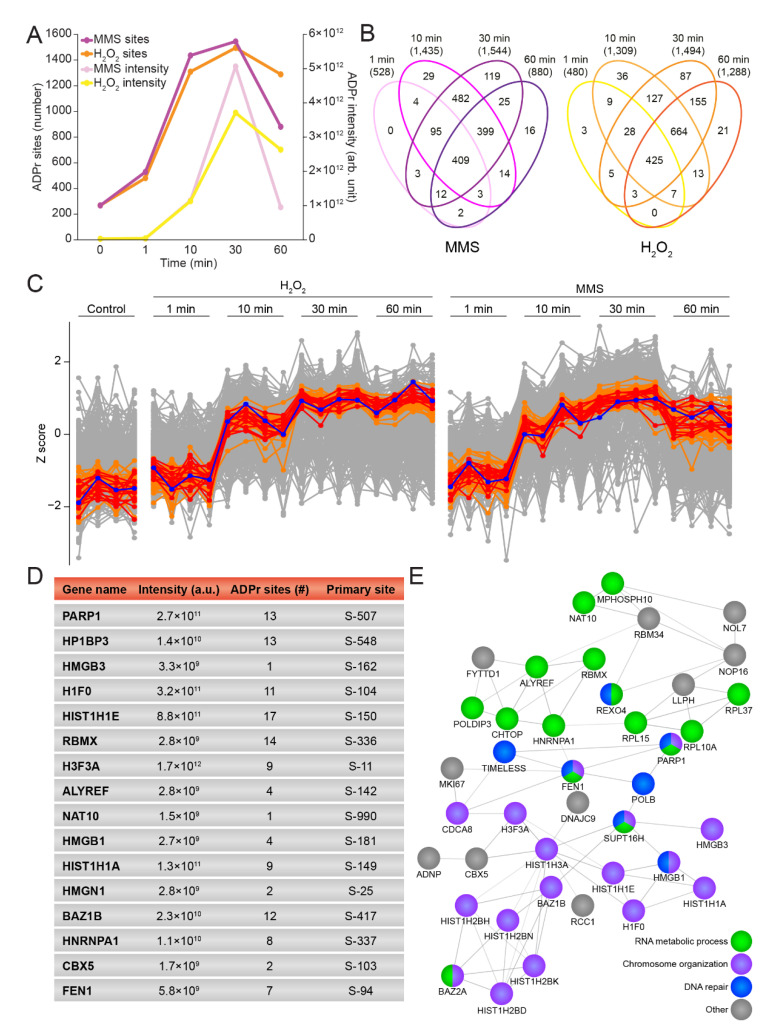
General temporal properties of the H_2_O_2_- and MMS-induced ADP-ribosylome. (**A**) Line graph showing the temporal effect on the total number of ADPr sites and the summed ADPr intensity for MMS and H_2_O_2_. Control condition corresponds to the 0 min time point. (**B**) Venn diagram depicting the number of identified and localized ADPr sites upon different treatment times with MMS (left panel) and H_2_O_2_ (right panel). (**C**) Profile cluster analysis illustrating the temporal properties of PARP1 (blue line) and the top 15 proteins (red lines) or top 50 proteins (orange lines) with the most similar profiles to PARP1. (**D**) Table including PARP1 and the 15 proteins showing the most similar profiles to PARP1. (**E**) STRING network showing interactions between the 50 proteins showing the most similar profiles to PARP1. Default STRING clustering was used (*p* > 0.4) except for disabling of text mining, and disconnected proteins were omitted from the network. Proteins were annotated with colors as highlighted in the figure legend.

**Figure 4 cells-10-02927-f004:**
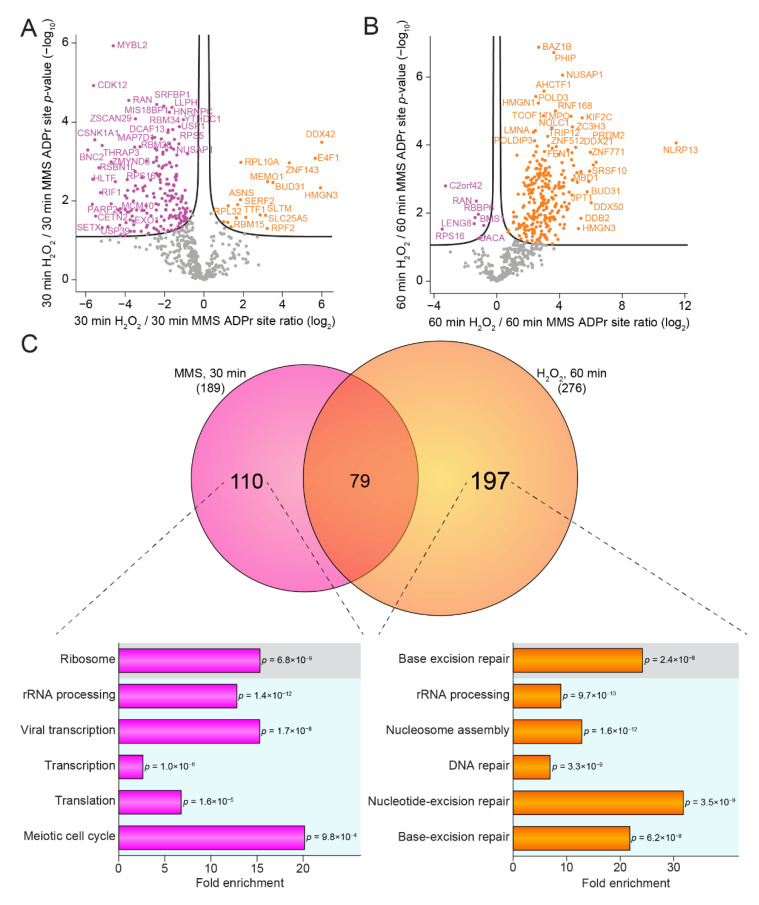
Temporal-specific changes in the ADP-ribosylome. (**A**) Volcano plot analysis visualizing the dynamics of ADPr target proteins after 30 min of H_2_O_2_ treatment compared to 30 min of MMS treatment. Significance was determined via two-tailed Student’s t-testing, with an FDR of 0.05, an s0 of 0.1, and 2500 rounds of randomization. Proteins significantly upregulated by MMS treatment are depicted in pink, proteins significantly upregulated upon H_2_O_2_ treatment are depicted in orange, and proteins not significantly regulated are shown in grey. (**B**) As A, but illustrating the changes after 60 min of treatment. (**C**) Scaled Venn diagram showing the overlap between proteins significantly upregulated upon 30 min of MMS treatment compared to proteins significantly upregulated upon 60 min of H_2_O_2_ treatment (top panel). Gene ontology enrichments of the proteins specific for those upregulated upon 30 min MMS compared to the total genome (bottom, left panel) and of the proteins specific for those upregulated upon 60 min of H_2_O_2_ compared to the total genome (bottom, right panel). Grey box: KEGG, blue box: GOBP.

## Data Availability

The mass spectrometry proteomics data have been deposited with the ProteomeXchange Consortium via the PRIDE [58] partner repository with the dataset identifier PXD028902.

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
