# Peer review of "Temporal and Site-Specific ADP-Ribosylation Dynamics upon Different Genotoxic Stresses"

_cells, 2021, doi:10.3390/cells10112927_

Round 1

Reviewer 1 Report

Article: Temporal and site-specific ADP-ribosylation dynamics upon different genotoxic stresses

Summary:

Sara C. Buch-Larsen et al present a paper ADP-ribosylation response after both MMS and H2O2 genotoxic treatment using mass spectrometry-based Af1521 enrichment and refined it in time, site- and protein-specific manner. Most of the data shows considerable overlap between both stress-inducing agents. Nevertheless, the fact this was done in parallel and summed up in one unique study, that authors further expand the stress-induced ADP-ribosylome and observed a shift over time in preferred ADP-ribosylated serine within PARP1 is, as authors say, “an important biological resource” .

Comments:

The paper is well written - clear and logical. Current version of the main manuscript contains poor-resolution images, and it was the biggest struggle while reading the manuscript. I guess it is not an actual mistake as the data is clear in the supplemental information. Still, I suggest some small changes that would make the existing text and figures clearer:

  1. I noticed two small mistakes in the text:

On page 1, row 31 - NAD+ should be changed to NAD+

On page 2, row 70 - methyl bitro-nitrosoguanidine should be changed to methyl nitro-nitrosoguanidine

  1. In Figure 1A (and S1A), it would be helpful to put arrows next to the bands corresponding to modified PARP1 and histones.

In Figure 1G, the S (for Serine) after K would be more obvious and intuitive than the blue star (or at least blue star should be mentioned in the figure legend).

In Figures 2E and 3E, the grey circles are not described in the legend (at least not defined/other would be helpful).

  1. Content of Figure 2A is interesting, but maybe would deserve a clearer representation as right now seems a bit confusing with all the potential residues depicted in the legend. Pie charts clearly show the dominance of Ser-ADPr but the change could be represented so it is more obvious. Maybe it will become clearer with the higher resolution, or authors could consider some other representation of their choice.

Reviewer 2 Report

This manuscript by Buch-Larsen and coworkers describes a study of the temporal evolution of ADP-ribosylated sites on proteins in cultured mammalian cells after chemically induced DNA damage, identified by MS after affinity enrichment using the binder Af1521. The study expands our knowledge of the damage induced ADP-ribosylome as it provides a direct comparison of the effects of two methods of DNA damage induction over time, and identifies considerably more target sites than previously identified. The authors find initial modification of mainly serines at multiple sites and a temporal transition to a higher degree of modification at fewer sites.

Taken together, the study appears technically sound (and provides excellent confirmation of the protocols developed by this group), the manuscript is well written and illustrated, and the conclusions drawn appear appropriate. I only have minor suggestions for revisions:

Line 233f, relative ADPr levels after MMS treatment in 2 cell lines: Formally, to draw this conclusion, a control experiment with relevant amounts of total proteins from both cell lines, treated and visualized on the same Western blot, should be executed. If this has been done it should be mentioned.

Introduction – Line 46ff, it would help less initiated readers to state briefly whether other residues (e.g., lysine) modification is considered a target residue for ADPr. “…when HPF1 is present”: perhaps more concise to say “when HPF1 is engaged”. Line 76, “unbiased method”, is this regarding the mentioned modifications (D, E, S) or regarding all possible modifications (including those detected and listed in Table S1)? Please consider revising.

Line 257, “collectively” – to me, this refers to both cell lines, both treatments across all time points. Otherwise, please consider re-phrasing, or save this for the description of Fig.2.

Line 325 ff, consider revising this sentence for clarity.
